# From Renewable Biomass to Water Purification Systems: Oil Palm Empty Fruit Bunch as Bio-Adsorbent for Domestic Wastewater Remediation and Methylene Blue Removal

**Cristina E. Almeida-Naranjo** [1,2,*], **Elvia Gallegos** [2,3], **Elizabeth Domínguez** [1], **Paola Gutiérrez** [1], **Vladimir Valle** [1,*], **Alex Darío Aguilar** [1,4], **Alexis Debut** [5] **and Catalina Vasco** [6]

1   Departamento de Ciencias de Alimentos y Biotecnología, Escuela Politécnica Nacional,
    Ladrón de Guevara E11-253, Quito P.O. Box 17-01-2759, Ecuador; elizabethdominguez8@outlook.es (E.D.);
    paola.gutierrez@epn.edu.ec (P.G.); alex.dario.aguilar@gmail.com (A.D.A.)
2   Faculty of Engineering and Sciences—Biotechnology Engineering, Universidad de las Américas,
    Redondel del Ciclista Antigua Vía a Nayón, Quito P.O. Box 17-01-24, Ecuador; elvia.gallegos@ute.edu.ec or
    elvia.gallegos@udla.edu.ec
3   Facultad de Ciencias, Ingeniería y Construcción, Universidad UTE, Rumipamba y Bourgeois,
    Quito P.O. Box 17-01-2746, Ecuador
4   Advanced Materials and Processes (MAP)—Technische Fakultät, Friedrich-Alexander-Universität
    Erlangen-Nürnberg, 91058 Erlangen, Germany
5   Center of Nanoscience and Nanotechnology, Universidad de las Fuerzas Armadas ESPE,
    Sangolquí P.O. Box 17-15-231B., Ecuador; apdebut@espe.edu.ec
6   Departamento de Ciencias Nucleares, Escuela Politécnica Nacional, Ladrón de Guevara E11-253,
    Quito P.O. Box 17-01-2759, Ecuador; catalina.vasco@epn.edu.ec
*   Correspondence: cristina.almeidan@epn.edu.ec or cristina.almeida@udla.edu.ec (C.E.A.-N.);
    vladimir.valle@epn.edu.ec (V.V.); Tel.: +593-2-2976-300 (C.E.A.-N. & V.V.)

**Abstract:** Oil palm empty fruit bunch fibers (OPEFBF), in three size ranges (small: 250–450 μm, medium: 450–600 μm, large: 600–800 μm), were investigated as a filter-bed material in biofilters for the removal of organic matter and nutrients. After saturation, these fibers (post) were used in the removal of methylene blue through batch processes. The batch adsorption tests included optimizing the adsorbent dosage (0.5–32.0 g/L) and contact time (2.5–60.0 min). Experimental data were fitted to various kinetic/isotherm models. Instrumental characterization of both raw and post fibers was conducted. Post fibers underwent morphological/compositional changes due to the presence of microorganisms and their byproducts. Efficiencies reached up to 94% for chemical oxygen demand (COD), 88.4% for total nitrogen and 77.2% for total phosphorus. In batch adsorption, methylene blue removal exceeded 90%, underscoring the effectiveness of small raw OPEFBF and large post OPEFBF. Kinetic models indicated that raw OPEFBF better fit the pseudo-first-order model, while post OPEFBF better fit the pseudo-second-order model. Both types of OPEFBF showed a good fit with the Freundlich model (higher $R^2$, lower $\chi^2$ and SSE). Particularly, large post OPEFBF stood out as the most efficient adsorbent, achieving a maximum adsorption capacity of 12.02 mg/g for methylene blue. Therefore, raw/post OPEFBF could be an alternative to remove contaminants from wastewater.

**Keywords:** post adsorbent; agro-industrial residues; biofilters; batch adsorption process; fiber size distribution

## 1. Introduction

Population growth, socioeconomic development and changes in the consumption model produce an increase in the global water demand (~3829 km³/year), as well as the generation of wastewater (~2212 km³/year) [1]. While there are no precise data available on wastewater treatment, there is a clear correlation with a country's income. High-income countries typically treat around 70% of their municipal and industrial wastewater, whereas low-income countries manage just 8% of it [2]. The low quantity of treated water

brings negative effects on the environment, human health, economy and water supply sustainability [3]. Therefore, treating wastewater is of the utmost importance.

Conventional approaches to sanitation and wastewater management, such as activated sludge systems and aerated lagoons, favor centralized collection and treatment. However, complex operation, energy consumption (0.3–0.4 kWh/m$^3$) and a high cost (i.e., 0.11 USD/m$^3$ conventional activated sludge) [4,5] have limited their application in rural or semi-urban territories. Decentralized approaches such as biofiltration technologies are favored in rural communities due to their low costs and efficiency [6].

Biofiltration technologies carry out biological, chemical and physical processes for contaminant removal. The filter-bed material plays an important role in these processes, allowing the development of the biota and the contaminant adsorption. The conventional filter materials used are granular activated carbon (GAC), ceramic, anthracite, gravel, sand, expanded clay and plastics, although they have some drawbacks, such as a high cost (e.g., GAC = 1.83 USD/kg) [7], clogging, longer start-up period and requirement of an external carbon source [8]. Thus, in recent years, interest in using low-cost adsorbents such as agro-industrial and industrial residues has increased. Agro-industrial residues have some advantages such as a high availability (2 billion tons/year) [9], good adsorption capacity (e.g., reactive dyes 179 mg/g and basic dyes 295 mg/g on coffee powder residues) [10], physicochemical stability, low or no cost (e.g., wood chips ~2.37 USD/m$^3$, wheat straws ~2.5 USD/m$^3$) [11] and being non-toxic and environmentally friendly [8]. Among the agro-industrial residues used in the removal of different contaminants are peanut shells (caffeine, organic matter) [12,13], woodchips (NO$_3$-N) [11], agave fibers (organic matter, pathogen microorganism) [14], oil palm (methylene blue: MB, crude oil, methyl orange dye) [15,16], etc.

Ecuador is the eleventh largest producer of African palm worldwide (470 metric tons/year) [17]. This industry has grown during the last few years along with its negative environmental impacts, such as effects on tropical forests, biodiversity and ecosystem services. Also, it generates around 1.72 million metric tons/year of solid residues, where oil palm empty fruit bunch fibers (OPEFBF) represent around 48% [18].

OPEFBF has been employed to adsorb dyes (e.g., reactive black 5, MB), metals, phenols and other contaminants from aqueous solutions through batch adsorption processes [19], achieving adsorption capacities between 7.34 and 50.76 mg/g [20], and 0.19 and 8.887 mg/g for dyes and metals. Raw OPEFBF adsorption capacity is associated with their surface area (1.48–28.4 m$^2$/g) and the presence of functional groups (e.g., hydroxyl, carboxylates, carbonyl, amides, phenol, alkyl) [21]. Moreover, OPEFBF biofilters removed 52% color, 49% turbidity, 44% TSS, 59% COD, 84% BOD and 94% NH$_3$-N in urban stormwater [22]. Oriausifo (2022) [23] used burnt oil palm kernel shells and sand for removing turbidity (<5 NTU). Furthermore, activated carbon from OPEFBF showed a higher adsorption for phenolic compounds than other parts of the oil palm tree [19].

One limitation of adsorption is that contaminants are only retained in the adsorbent and they are not degraded [24]. Consequently, the subsequent treatment and/or disposal of these spent or post adsorbents raise notable environmental concerns, including the potential permeabilization of surrounding soil, groundwater and surface water through natural leaching or desorption mechanisms [25]. Several efforts have been made to manage this issue, including soil amendment, capacitor utilization, catalyst/catalyst support application, advanced oxidation processes, cement production, ceramic production and secondary adsorption [24,26,27]. The final approach aims to maximize the potential of spent sorbents without the need for additional reagents for regeneration. For instance, wheat straw has been successfully reused to adsorb congo red after neutral red (neutral red adsorption capacity 6.25 mg/g, neutral red-loaded 49.7 mg/g) from aqueous solution in a fixed-bed column. Another study reused walnut shell to remove congo red after methyl blue from aqueous solutions, yielding favorable outcomes (methyl blue adsorption capacity = 0.0039 mmol/g, MB-loaded adsorption capacity = 0.03–0.041 mmol/g) [28].

Although the utilization of OPEFBF for the removal of contaminants from wastewater through filtration processes has been reported in the literature, there remains a significant knowledge gap regarding the influence of fiber size and filter bed height on the efficacy of contaminant removal. Furthermore, limited attention has been given to investigating the potential for reusing OPEFBF as a post-adsorbent material. Thus, this study aims to address these research gaps by systematically evaluating the impact of both fiber size and filter-bed height on the removal efficiency of organic matter and nutrients (N, P) from synthetic domestic wastewater. Additionally, the adsorption capacity of OPEFBF (saturated) for MB will be assessed after the adsorption processes.

## 2. Materials and Methods

### 2.1. Wastewater Characterization

The synthetic domestic wastewater was prepared based on the composition defined by Almeida Naranjo et al. (2017) [29]. It was prepared daily to preserve its characteristics. The physicochemical characteristics are described in Table 1. These properties emulate a medium intensity real domestic wastewater [30].

**Table 1.** Physicochemical characteristics of synthetic wastewater.

| Parameter | Unit | Average Value | Standard Deviation |
|---|---|---|---|
| Chemical Oxygen Demand | mg/L | 589.71 | 46.54 |
| Total Nitrogen | mg/L | 21.60 | 0.48 |
| Total Phosphorous | mg/L | 12.14 | 0.37 |
| Volatile Solids | mg/L | 517.20 | 39.46 |
| Total Solids | mg/L | 111.05 | 40.14 |
| Temperature | °C | 17.40 | 0.61 |
| pH | - | 6.48 | 0.35 |
| Dissolved Oxygen | mg/L | 6.75 | 0.22 |

### 2.2. Oil Palm Empty Fruit Bunch Wastes Conditioning

Oil palm empty fruit bunch (OPEFB) residues were collected from oil palm industries ubicated at Quinindé, Ecuador. Then, OPEFB were cleaned, and the fibers were separated manually from the central peduncle, pre-dried at room conditions for 3 days, and dried in a MMM Group model VENTICELL LSIS-B2V/VC 55 oven at 105 °C for 5 h. Afterwards, OPEFBF were ground and sieved in three size ranges: 250–425 µm (small), 425–600 µm (medium) and 600–850 µm (large) [31]. Obtaining raw OPEFBF, the length of the raw OPEFBF was determined using the methodology described by Aguilar et al. (2022) [32], resulting in approximately 20,000 data points for each fiber size. The conditioned OPEFBF were stored in polypropylene bags with hermetic closure at room temperature to prevent moisture.

### 2.3. Experimental Model

#### 2.3.1. Biofilters Operation

The pilot scale unit consisted of six biofilters built with polyvinyl chloride tubes (Øint = 8.1 cm) (Figure 1). The biofilters were configured with 3 layers, described from bottom to top as follows: a 10 cm support layer (gravel, diameter = 10–25 mm), an active/middle layer (raw OPEFBF: small, medium, large) with two different heights (60 and 90 cm) and a 5 cm top layer (gravel, diameter = 5–8 mm). Biofilters were inoculated with municipal wastewater from Quito city (around 3 L) and conditioned with synthetic domestic wastewater (5 mL/min) until reaching a pseudo-stationary state (around 3 weeks). Biofilters operated intermittently for 8 h/day, 5 days/week for 17 weeks with a hydraulic load rate of 0.5 $m^3/m^2d$ [12]. The wastewater feed was carried out through a 100 L elevated tank.

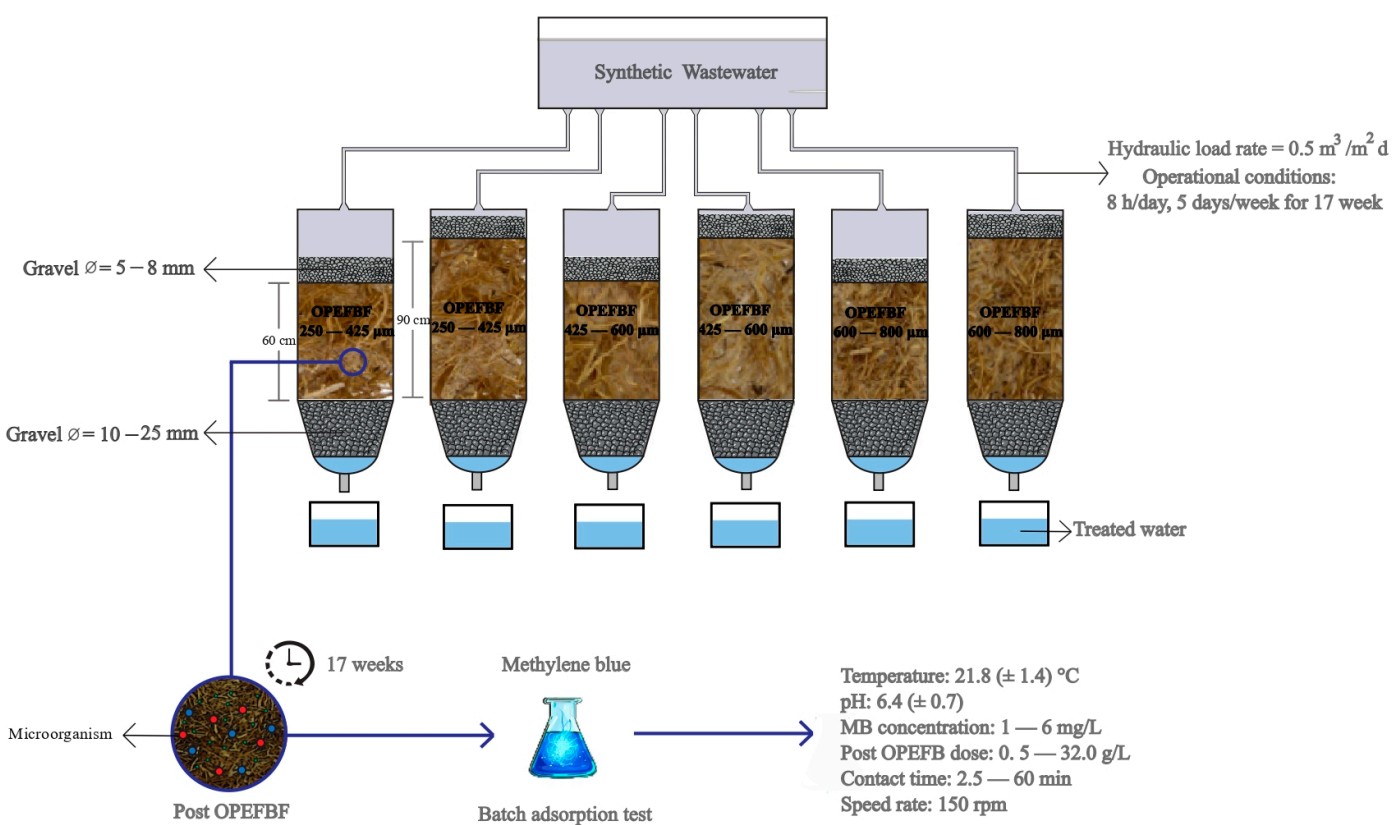

**Figure 1.** Experimental model.

### 2.3.2. Batch Adsorption Tests Using Post Adsorbents

After 17 weeks of biofilter operation, the saturated OPEFBF (post OPEFBF) were removed from the filters and dried. Raw and post OPEFBF were used to remove MB (CAS 7220-79-3, Sigma Aldrich Inc., ≤100%, St. Louis, MO, USA). Synthetic solutions of 4 mg/L of MB were used in the batch adsorption tests. The solutions (volume V = 30 mL) were placed into Erlenmeyer flasks and mixed on an orbital shaking table at 150 rpm. Optimal doses of raw and post OPEFBF were determined using different doses between 0.5 and 32.0 g/L, for 60 min. The contact time and adsorption kinetics were determined using the optimal OPEFBF dose for each fiber size. Contact times between 2.5 and 60.0 min were tested for the MB adsorption. Adsorption isotherms essays were performed using optimal dose/contact time at 6 concentrations (1–6 mg/L) of MB. All batch adsorption tests were performed in triplicate for the three fiber sizes, keeping pH = 6.4 (±0.7) and room temperature = 21.8 (±1.4) °C. The efficiency in the MB removal was analyzed in batch adsorption tests.

### 2.4. *Analytical Methods*

### 2.4.1. Influent/Effluent Monitoring

Influent and effluent were characterized every week. Hach methods were used for COD, total nitrogen (TN) and total phosphorus (TP). COD was measured by the 8000 method (Reactor Digestion Method—HR: 20 to 1500 mg/L), TN was determined by the 10071 method (Persulfate Digestion Method—LR: 0.5 to 25.0 mg/LN), and TP was established by the 10127 method (Molybdovanadate Method with Acid Persulfate Digestion1—HR: 1.0 to 100.0 mg/L $PO_4^{3-}$). The total volatile solids (TVS) and total suspended solids (TSS) were quantified following APHA-AWWA-WPCF (2017) [33]. TVS and TSS were measured using the method 2540 E (Fixed and Volatile Solids Ignited at 550 °C) and the method 2540 D (Total Suspended Solids Dried at 103–105 °C), respectively. In addition, once a week, an HQ30d portable multiparametric was used to measure pH,

temperature (T) and dissolved oxygen (DO). COD was measured twice a week, and the rest of the parameters were measured once a week.

### 2.4.2. Raw/Post OPEFBF Characterization

Raw OPEFBF were characterized by moisture [34], extractives [35], ash (method: D1102-21), lignin, cellulose and hemicellulose [36,37], according to ASTM standards.

To understand the influence of raw/post OPEFBFs' composition and structure on their adsorption capacity, thermogravimetric analysis (TGA) was performed using a METTLER TOLEDO thermo-balance model TGA-2, covering a temperature range of 25 to 750 °C with a heating rate of 10 °C/min and a nitrogen flow of 50 mL/min. Functional groups' identification was carried out using a JASCO FT/IR-C800 spectrometer. Twenty-five scans were conducted in the range of 4000 to 400 cm$^{-1}$ with a resolution of 2 cm$^{-1}$. SEM analysis was performed on a FEI Phenom model FP3950/00an electron microscope. OPEFBF fibers were coated with Au/Pd in an argon atmosphere for 135 s under a current of 18 mA using a Sputter Coater SC7620 (Quorum). Finally, the specific surface area (Brunauer-Emmett-Teller, BET) of the small raw OPEFBF was determined by nitrogen adsorption in a micrometric NOVA touch 1LX equipment. More than 6 multi-points were considered to determine the surface characteristics of the material, which was conditioned by drying it at 105 °C under vacuum.

### 2.4.3. MB Quantification before/after the Adsorption Batch Tests

The wavelength of maximum absorption of MB (664 nm) was determined by scanning between 200 and 800 nm using a 6 mg/L solution [38]. The calibration curve of MB (y = 0.1988x − 0.012, R$^2$ = 0.997) was constructed using solutions with concentrations between 0.1 and 6.0 mg/L [21]. The concentration of MB before/after the adsorption processes was performed in a Analyticjena Specord$^®$ 210 Plus UV-VIS spectrophotometer, Analytik, Jena, Germany.

### *2.5. Data Analysis*
### Isotherm and Kinetic Models

The data obtained in the kinetics tests were fit to the non-linear pseudo-first-order (Equation (1)), pseudo-second order (Equation (2)), and Elovich (Equation (3)) models:

$$q_t = q_e \left( 1 - e^{k_1 t} \right) \tag{1}$$

$$\frac{t}{q_t} = \frac{1}{k_2 q_e{}^2} + \frac{t}{q_e} \tag{2}$$

$$q_t = \frac{1}{\beta} \ln(1 + \alpha\beta t) \tag{3}$$

Moreover, to obtain information about the adsorption process, the intraparticle diffusion model was applied (Equation (4)).

$$q_t = k_p \sqrt{t} + C \tag{4}$$

where $q_t$ (mg/g) is the amount of MB adsorbed at time t, $q_e$ (mg/g) is the amount of MB adsorbed at equilibrium, $k_1$ (min$^{-1}$) and $k_2$ (g/(mg min)) are the pseudo-first-order and the pseudo-second-order rate constant, respectively. $\alpha$ (mg/g min) is the initial rate constant, $\beta$ (mg/g) is the desorption constant, $k_p$ (mg/g min$^{1/2}$) is the rate constant of the intra-particle diffusion model and C (mg/g) is a constant associated with the thickness of the boundary layer [39].

The data obtained in the isotherm tests were fitted to the non-linear models of Langmuir (Equation (5)), and Freundlich (Equation (6)) and Sips (Equation (7)), which [40]:

$$q_e = \frac{q_m K_L C_e}{1 + K_L C_e} \tag{5}$$

$$q_e = K_F C_e^{1/n} \tag{6}$$

where $q_e$ (mg/g) is the amount of MB adsorbed per unit mass of OPEFBF at equilibrium, $q_m$ (mg/g) is the maximum adsorption capacity of OPEFBF, $C_e$ (mg/L) is the liquid-phase concentration of MB in equilibrium, $K_F$ (mg/g) is the Freundlich capacity constant, $K_L$ (L/mg) is the Langmuir constant and n is the Freundlich intensity parameter [41]. The applied kinetics/equilibrium models have shown a good fit ($R^2 \sim 1$) when agro-industrial residues were used in the removal of different contaminants [40].

The adsorption tests were performed in triplicate, and the control was placed with distilled water. Furthermore, the adsorption tests were carried out in the dark to avoid photodegradation.

### 2.6. Statistical Analysis

#### 2.6.1. Biofilters

The two-way analysis of variance (two-way ANOVA) was used to analyze the performance of the biofilters (COD, TN, TP, TSS, TVS), as well as to determine design factors' influence (filter-bed depth and fiber size). Differences were considered significant at $p \leq 0.05$.

#### 2.6.2. Batch Adsorption Tests

The optimal dose of raw/post OPEFBF in the different sizes was determined by means of the significant differences between the doses used in the adsorption of MB and the percentage of removal efficiency achieved in batch tests. One-way analyses of variance (one-way ANOVA) were used. Also, Tukey's test with a significance level of 0.05, was applied.

The statistical analysis of data from kinetics and isotherm models, considered the determination of means, standard deviation, error, and linear regressions. For this purpose, in batch and continuous tests, the coefficient of determination ($R^2$) and the chi-square ($\chi^2$) were calculated (Equations (7) and (8)) to determine the models that best fit the caffeine and triclosan adsorption data:

$$R^2 = 1 - \frac{\sum (V_{e,exp} - V_{e,cal})^2}{\sum (V_{e,exp} - V_{e,mean})^2} \tag{7}$$

$$\chi^2 = \sum \frac{(V_{e,exp} - V_{e,cal})^2}{V_{e,cal}} \tag{8}$$

where $V_{e,exp}$ are the experimental value of parameters (adsorption capacity, initial/final MB concentrations for batch tests and fixed-bed columns, respectively), $V_{e,cal}$ are the calculated parameters using the Solver tool, and $V_{e,mean}$ is the mean of $V_{e,exp}$ values [39].

### 3. Results

### 3.1. Raw/Post OPEFBF Characterization

The fiber sizes followed a gamma distribution, with the following structural similarities: large to medium at 79.07%, large to small at 69.72% and medium to large at 90.08%.

Figure 2 shows the FTIR spectra of raw/post OPEFBF. In raw OPEFBF, the presence of hydroxyl groups (~3300 and 1650 cm$^{-1}$), methyl (~2975 and 2890 cm$^{-1}$), methylene (~2920 and 2850 cm$^{-1}$), carbonyl (~1700 cm$^{-1}$), acetal/ketal (glycosidic bond~1159 and 1031 cm$^{-1}$)

and aromatic groups (~1667–1450, 1244, 667 cm$^{-1}$) was determined. These groups show the presence of water, lignin, carboxylic acids, hemicellulose and cellulose, characteristic constituents of OPEFBF. Furthermore, in cellulose the presence of the amorphous phase (~826 cm$^{-1}$) and the crystalline phase (~1420 cm$^{-1}$) was observed [32], while the post OPEFBF in its three sizes present qualitative changes in the intensity of different bands with respect to the OPEFBF. We noticed decreases in the methylene group (between 2927 and 2848 cm$^{-1}$) and increases in the bands of the methyl (between 2977 and 2886 cm$^{-1}$ and 1375 and 1372 cm$^{-1}$), hydroxyl (between 3329 and 3290 cm$^{-1}$ and 1650 and 1629 cm$^{-1}$), ketal/acetal (between 1031 and 1028 cm$^{-1}$) and aromatic groups (Ar-R, C=C) in the range of 1244 to 1237 cm$^{-1}$. These changes are associated with the microbial activity that took place in the raw OPEFBF during the operation of the biofilters [42]. Large particles (600–850 μm) promoted yeast growth (Figure 2) because they allowed greater oxygenation and availability of space so that these organisms could carry out their metabolic processes effectively [43]. Yeasts can break the fatty acid chains present in raw OPEFBF, because their peroxisome contains the enzymes necessary to carry out their β-oxidation [42]. Likewise, yeasts use cellulose, hemicellulose and lignin as carbon sources for their metabolism. In metabolism, the hydrolysis of cellulose occurs where enzymes of the glucoside hydrolase type intervene. β-glucosidases break the bond of the disaccharide cellobiose in two glucose molecules, but first it is necessary for other enzymes to act such as endo-β-1,4-glucanases that internally break the cellulose chain, fragmenting it, while exoglucanases separate simple sugars from the ends [44]. Now, the breakdown of glucose causes it to follow the metabolic pathway of glycolysis, obtaining two molecules of pyruvate, which are then converted into acetyl-CoA. These same molecules can be converted into malonyl-CoA. Malonyl-CoA is composed of two carbon molecules that are used to initiate the synthesis of carboxylic acids. Palmitic acid is the final product [42,44], which is evidenced by the increase in the qualitative intensity of the bands of the carbonyl group included in the range of 1732 to 1727 cm$^{-1}$ [32]. On the other hand, a larger particle size negatively affects the growth kinetics of the bacteria; this is because it will take longer for simple sugars to be produced, that is, the substrate will be consumed slowly. These types of microorganisms prefer a smaller particle size (larger surface area) in their food. However, bacteria consume less than they produce, so their metabolic residues (biofilm, Figure 2) accumulate. These residues are mainly formed by sugars and proteins, the same ones that produce the change presented in the FTIR spectrum [43].

The TGA/DTG results of OPEFBF are presented in Figure 2d. An initial mass loss below 150 °C (volatile material) was observed in both raw and post OPEFBF, attributed to water evaporation. Between 150 and 410 °C (moderately volatile material), a degradation process with a maximum decomposition temperature at approximately 310 °C was noted. This is associated with the degradation of hemicellulose, cellulose and lignin, involving the removal of polyhydroxyl groups, accompanied by the depolymerization and decomposition of macromolecules, resulting in the formation of b-(1,6) anhydrous D-glucopyranose (levoglucosan), 2-furaldehyde (furfural) and volatile products of lower molecular weight. Finally, at temperatures above 410 °C, a residue rich in carbon and minerals remains [45,46]. Likewise, according to the DTG analysis, during microbial degradation, primarily of cellulose, hemicellulose and lignin, compounds such as organic acids, aldehydes, ketones, partially degraded polysaccharides, phenolic compounds and nitrogen-containing compounds were generated [42,43]. The structures of these compounds can act as inhibitors for thermal reactions by altering the chemical balance or the availability of reaction sites. This interference could lead to a decrease in kinetics, reflected in a second decomposition temperature at around 360 °C [47].

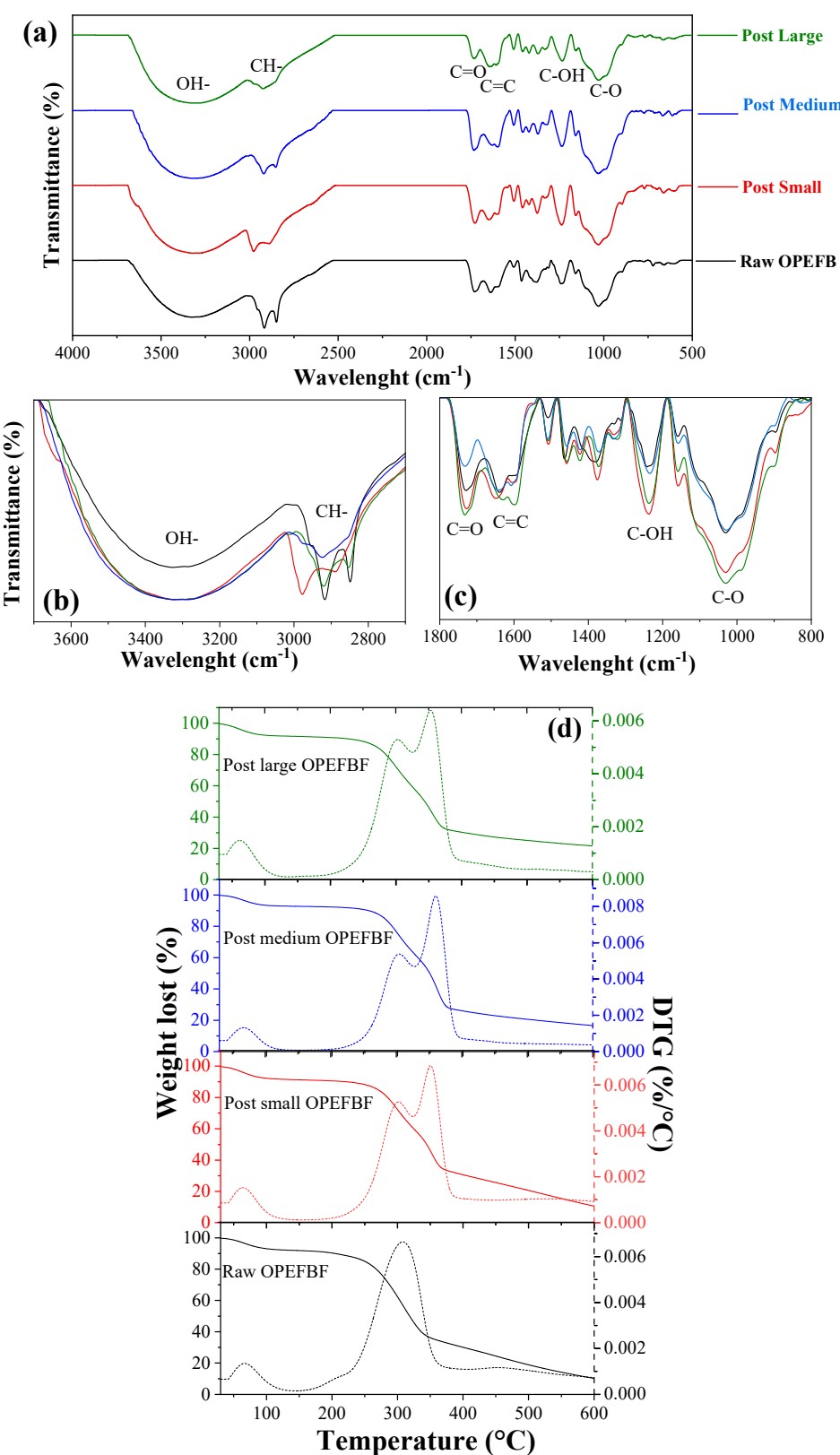

**Figure 2.** FTIR spectra and TGA of raw/post OPEFBF. (**a**) FTIR spectra between 500 and 4000 nm, (**b**) FTIR spectra: zone expansion 3700–2700 cm$^{-1}$, (**c**) FTIR spectra: zone expansion 1800–800 cm$^{-1}$, (**d**) TGA of raw/post OPEFBF (TGA = solid line, DTGA = dotted line).

Figure 3 demonstrates that the raw OPEFBF possess an irregular surface, accompanied by low cavities and pores. This was confirmed by the BET analysis result, which indicates

a low surface area (0.47 m$^2$/g). Meanwhile the SEM images from the post OPEFBF showed morphological changes and the presence of microorganisms. The small post OPEFBF shows the presence of biofilm with few yeasts, without major damage to the surface of the fiber, which suggests that the biofilm acts as protection for OPEFBF. As the particle size increases, a greater number of yeasts and greater effects on the surface of the fiber (greater porosity and cavities) are observed. The post large OPEFBF were the ones that presented the greatest morphological changes; the yeasts were even located within these irregularities and this is associated with the advantages that this particle size would have for these unicellular fungi [42]. Therefore, the chemical (functional groups) and irregular surface characteristics of raw/post OPEFBF suggest that they may have good adsorbent properties.

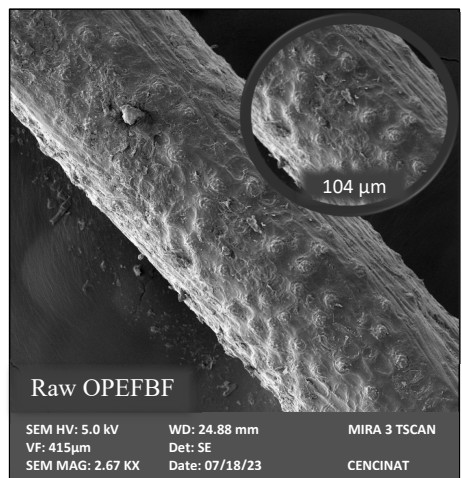
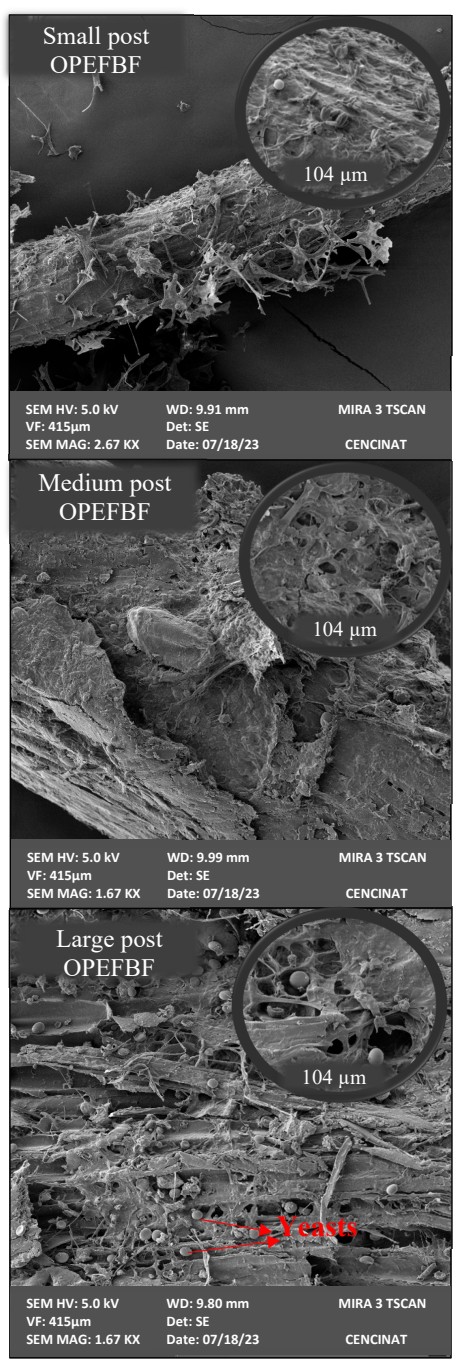

**Figure 3.** SEM images of raw and post OPEFBF.

### 3.2. Biofilter Performance

The average temperature (16.5–20.3 °C), pH (6.19–6.51) and dissolved oxygen (4.18–6.32 mg/L) of treated water were monitored every week. The measurements of the number of the flow rate and hydraulic rate were 3.71 ± 0.51 mL/min and 0.48 ± 0.08 $m^3/m^2d$, respectively. Figure 4 shows the biofilters' performance. B2 was the most efficient in removing COD (90.9%). This was related to the reduced dimensions of the fiber (250–425 μm), which in turn gives it a greater surface area, and allows it to effectively trap organic contaminants. Furthermore, the greater depth of the bed helps to prolong the retention time of wastewater and improve its treatment. The biofilters, except B4 and B6, show efficiencies that exceed 85.0%. B4 (425–600 μm) and B6 (600–850 μm), height = 90 cm, in Weeks 13 and 15, showing fluctuations in COD removal, which can be associated with the presence of yeasts (Figures 2 and 3). Moreover, a greater bed depth increased the contact time between the wastewater and the fiber modified by the microorganisms, which resulted in a higher COD concentration due to the degradation suffered by the material, which led to a lower efficiency of removal [48]. Likewise, it is verified that the biofilm formed in the smallest OPEFBF (Figure 3) acts as a protection for the fiber. The efficiency of biofilters is comparable to other studies with conventional biofilters (COD removal of 74.0–94.0%), and biofilters with organic media: rice straw, wood chips of orange tree, date palm fiber, peanut shell (COD removal 37.0–84.0%) [8].

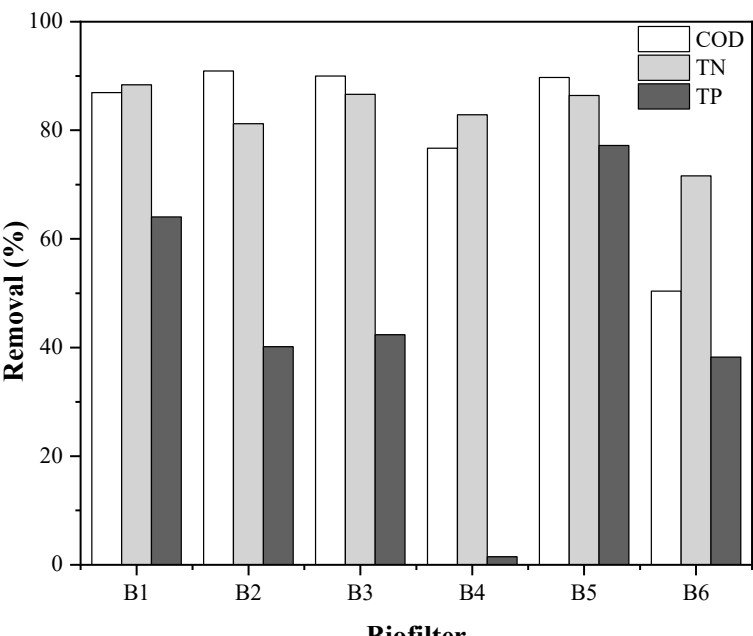

**Figure 4.** Biofilter performance. B1 = small OPEFBF, OPEFBF height = 60 cm, B2 = small OPEFBF, OPEFBF height = 90 cm, B3 = medium OPEFBF, OPEFBF height = 60 cm, B4 = medium OPEFBF, OPEFBF height = 90 cm, B5 = Large OPEFBF, OPEFBF height = 60 cm, B6 = large OPEFBF, OPEFBF height = 90 cm.

Regarding TN removal efficiency, B1 is the most efficient (88.4%), followed by B3 and B5. Lignocellulosic surfaces facilitate a robust interaction between ammonium ions and the anionic surface groups of the fiber: carboxylic, phenolic, hydroxylic [49]. For instance, surface carboxylic groups promote a negative charge, enabling the adsorption of cationic ions such as $NH_4^+$ onto its surface [50]. Simultaneous nitrification–denitrification was performed in the biofilm, creating anoxic microzones due to limited oxygen penetration. Denitrifying bacteria in the lower layer utilized nitrate produced by nitrifying bacteria in the outer layer [51]. Ammonia-oxidizing bacteria are located at the top of biofilters due to their accessibility to nutrients such as phosphorus, which diminishes as wastewater percolates through [52]. Hence, 0.6 m biofilters demonstrate a greater efficiency as nitrifying

bacteria inhibit the growth of fungus and yeast [53]. Some studies show that the TN removal efficiency achieved using different organic media (e.g., wheat straw, wood chips, Brewer's spent grains, etc.) has reached approximately 70.0% [8]. Therefore, using OPEFBF for TN removal is a preferable option.

B5 exhibited the highest TP removal efficiency (77.2%), followed by B1. Notably, shorter biofilters demonstrated enhanced phosphorus removal, attributed to the inhibitory effect of fungal colonies on the contact between synthetic wastewater and OPEFBF [54]. In comparison to TN removal, TP removal was relatively lower due to the system's primary focus on nitrification, which limited the biological removal of phosphorus in the presence of nitrates. The pH of the aqueous solution also played a crucial role, with a higher pH (>4.5) favoring cation adsorption and a lower pH (1.5–4) favoring anion adsorption [55]. In prior investigations on date palm fiber, it was demonstrated that TP removal efficiency reaches approximately 50% [8]. Nevertheless, the present study has exceeded the mentioned value.

Referring to TVS, the most efficient biofilter was B3 (83.0%), followed by B5, B1 and B2. Related to TSS, B1 was the first one to achieve a complete remotion in the 11th week, due to the packaging material being more porous and the path through which the wastewater passed being sufficient to retain all the TSS. Usually, a greater depth of the filter bed increases the TSS retention, and as a consequence the removal efficiency improves, although it should be noted that the highest solids retention occurs in the upper 0.1 m [48]; therefore, 0.6 m is an adequate filter depth. All biofilters reached 100.0% of remotion since the 14th week, which means that OPEFBF represent an excellent remotion material. The results are in accordance with the studies conducted by [56,57].

Previous research [56] highlighted that a smaller fiber size generally provided a larger surface area for adsorption (more activated sites). Considering similarities in fiber size distribution (>69.0%), there was no statistically significant difference between the three OPEFBF sizes. In contrast, bed height showed significant results. Tejada-Tovar et al. (2018) [55] indicate that as the bed height increases, the number of active sites also increases, resulting in improved organic matter and nutrient removal. However, it should be noted that the highest height is not necessarily the most effective, especially when there are microorganisms that can modify the filter medium [58]. Therefore, further exploration with a wider range of fiber sizes and bed heights is recommended, and the use of selectively modified fibers could provide valuable insights for future studies. Finally, there was no significant interaction observed between the two parameters.

### 3.3. Batch Adsorption Tests

#### 3.3.1. Optimal Dose and Contact Time

Optimal adsorption conditions are showed in Figure 5. It is observed that an increase in the dose of OPEFBF, in three particle sizes, favors the adsorption of MB. This is because an increase in the adsorbent dose allows a greater number of sites available for the MB adsorption [59]. In the case of raw OPEFBF, the smallest particles (250–425 μm) present the lowest dose (2 g/L) and contact time (20 min), reaching efficiencies of 93.1 ± 0.6%. Medium OPEFBF (425–600 μm) and large OPEFBF (600–850 μm) required a dose of 4 g/L, with optimal contact times of 20 and 30 min, respectively. With these two sizes, removal efficiencies of 87.3 ± 0.7 and 88.7 ± 0.8% were achieved. The greater efficiency of the smaller fibers, as with COD, is associated with the larger surface area (and thus a greater number of active groups) present in the raw OPEFBF [21]. The efficiency of medium and large OPEFBF did not present significant differences ($p$ value > 0.05); this is because they present a very high similarity in distribution in terms of their size similarity (>90%).

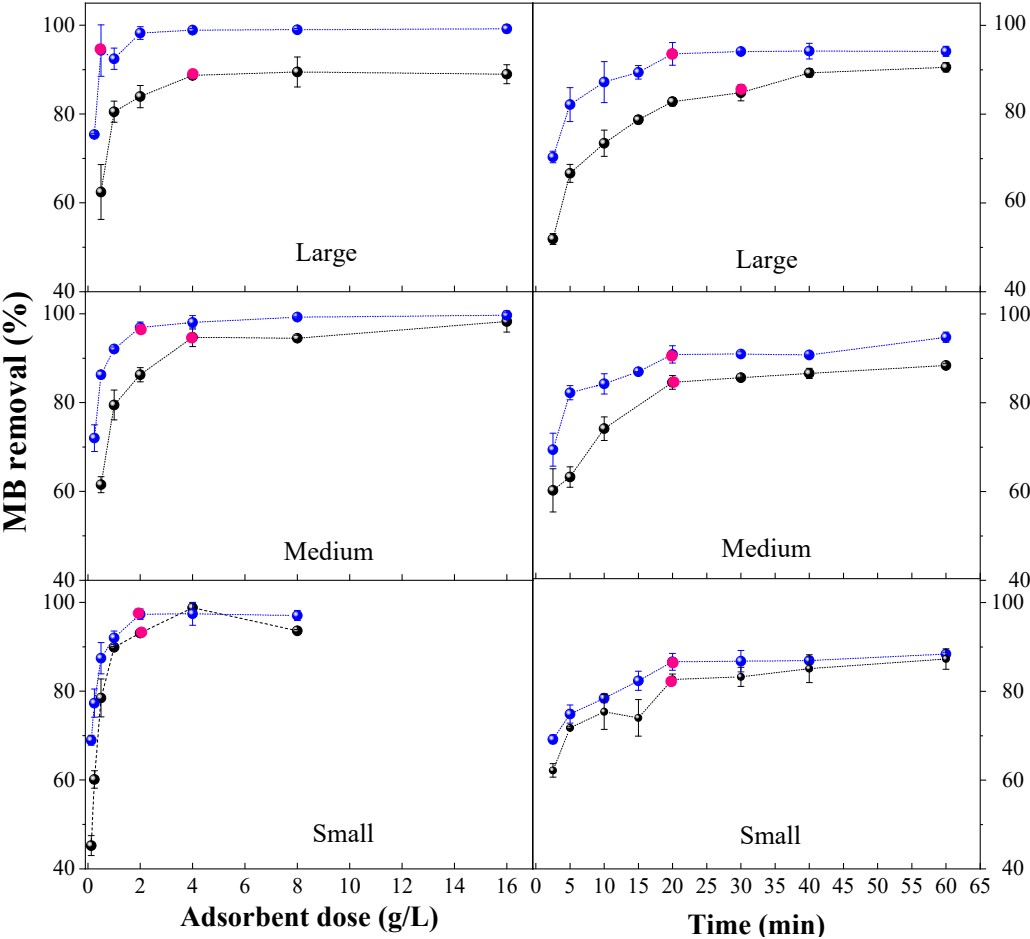

**Figure 5.** Optimal dose and contact time for MB removal using raw/post OPEFBF. Fuchsia points = optimal dose/contact time values for the removal of MB.

Regarding the contact time (Figure 5), it is observed that the raw OPEFBF achieve the highest MB removal in the first 10 min, reaching efficiencies of 86.9 $\pm$ 1.4% (small OPEFBF), 75.6 $\pm$ 0.2% (medium OPEFBF) and 73.5 $\pm$ 3.0% (large OPEFBF). This is attributed to the high availability of active sites in the OPEFBF. However, with the reduction in the accessible surface area to the solvent, the adsorbent reaches its saturation point, and adsorption primarily occurs through inclusion. As the reaction time increases, the efficiency of adsorption decreases due to the decreased internal surface area. It can be said that the slight increase in adsorption capacity over time is due to the growing difficulty in occupying vacant active sites due to the repulsive effects between MB molecules and the adsorbed molecules [21].

Although the optimal dose and contact time for the small and medium post OPEFBF are the same and there are no significant differences in the removal of MB, the removal efficiencies were higher: 97.4 $\pm$ 1.2% and 90.8 $\pm$ 0.9%, respectively. Something similar happens with contact time. While the post large ones do present a significant improvement (*p* value $\leq$ 0.05) in their MB removal efficiency, the dose and contact time were reduced 8 and 2 times, respectively. The improvements in the efficiency of post large fibers are produced by the action of microorganisms or their enzymes in the raw OPEFBF. The enzymes produced by fungi increase the number of pores/cavities in the adsorbent (Figure 3) as a product of the degradation of complex compounds of the cell wall (e.g., lignin, cellulose, hemicellulose). In addition, the enzymes that remain in the OPEFBF (e.g., laccase, xylanase, cellulase, lignin peroxidase) activate the surface of the fibers and this favors the removal of dyes. A similar behavior was obtained with banana peel treated for 9 weeks with fungal enzymes [60].

### 3.3.2. Adsorption Kinetics

In the adsorption process, kinetics plays an important role as it provides vital information about the reaction pathway and the mechanism controlling the rate of reactions [61]. Table 2 and Figure 6 show the fit of the MB removal kinetics data in the raw/post OPEFBF, where a good fit to the kinetic models used is observed ($R^2$ = 0.91–0.999). However, the adsorption of MB in the small and medium raw OPEFBF fit better to the pseudo-first-order model ($R^2$ = 0.980–0.999), while the raw large OPEFBF fit better to the pseudo-second-order model ($R^2$ = 0.996), although the value of $q_e$ obtained with the pseudo-first-order model was closer to the experimental $q_{e\,exp}$. For post OPEFBF samples of all sizes, they exhibit a superior fit to the pseudo-second-order model. The $\chi^2$ and SSE values show the lowest values (close to zero) for the models that best fit the data obtained (larger $R^2$, closer to one). The pseudo-first-order kinetic model (proposed by Lagergren in 1898) describes the removal process as adsorption preceded by diffusion through a boundary. The pseudo-first-order equation assumes the adsorption of an adsorbate molecule onto an active site on the adsorbent's surface [61,62]. On the other hand, the best fitting provided by the pseudo-second-order model indicates that the rate-limiting step for MB adsorption in OPEFBF is likely chemisorption, involving valence forces and electron transfer between the adsorbent and the adsorbate. This suggests the possibility that two adsorption sites on the adsorbent's surface could be occupied by a divalent dye ion, possibly due to the formation of $(MB^+)_2$ or $MBH^{2+}$ in an aqueous solution [61,63]. The adsorption of MB also followed a pseudo-second-order model when using fallen coconut leaves (16.08 mg/g) [64], citrus limetta peel (23.33 mg/g), cotton stem (104.82 mg/g), husk ash of rice (23.49 mg/g) [65] and *Terminalia catappa* husk (2.54 mg/g) [66].

**Table 2.** Parameters of kinetics models for MB adsorption.

| Model Type | Parameter (Units) | Small OPEFBF | | Medium OPEFBF | | Large OPEFBF | |
|---|---|---|---|---|---|---|---|
| | | Raw | Post | Raw | Post | Raw | Post |
| **Pseudo-first order** | $q_{e\,exp.}$ (mg/g) | 1.420 | 2.052 | 0.969 | 1.544 | 0.926 | 6.329 |
| | $q_e$ (mg/g) | 1.460 | 2.041 | 0.968 | 1.511 | 0.941 | 6.215 |
| | $k_1$ (min$^{-1}$) | 0.194 | 0.414 | 0.641 | 0.017 | 0.333 | 0.542 |
| | $R^2$ | 0.980 | 0.992 | 0.999 | 0.994 | 0.973 | 0.992 |
| | $\chi^2$ | 0.006 | 0.005 | $8.001 \times 10^{-5}$ | 0.002 | 0.003 | 0.039 |
| | SSE | 0.045 | 0.028 | $4.801 \times 10^{-4}$ | 0.012 | 0.022 | 0.275 |
| **Pseudo-second order** | $q_e$ (mg/g) | 1.650 | 2.157 | 0.998 | 1.574 | 1.024 | 6.510 |
| | $k_2$ (g/(mg min)) | 0.152 | 0.347 | 1.616 | 0.804 | 0.487 | 0.168 |
| | $R^2$ | 0.964 | 0.999 | 0.998 | 0.998 | 0.996 | 0.999 |
| | $\chi^2$ | 0.012 | $5.018 \times 10^{-4}$ | $3.225 \times 10^{-4}$ | $5.896 \times 10^{-4}$ | $4.334 \times 10^{-4}$ | 0.004 |
| | SSE | 0.081 | 0.003 | 0.002 | 0.004 | 0.003 | 0.030 |
| **Elovich** | $\alpha$ (mg/(g min)) | 1.201 | 108.673 | 163.297 | 651.832 | 5.710 | 556.341 |
| | $\beta$ (mg/g) | 3.345 | 0.472 | 0.189 | 0.943 | 7.493 | 0.202 |
| | $R^2$ | 0.918 | 0.986 | 0.989 | 0.990 | 0.994 | 0.992 |
| | $\chi^2$ | 0.026 | 0.008 | 0.001 | 0.003 | $4.334 \times 10^{-4}$ | 0.041 |
| | SSE | 0.184 | 0.050 | 0.009 | 0.002 | 0.005 | 0.284 |
| **Intra particle diffusion** | $k_{p1}$ (mg/(g min$^{1/2}$)) | 0.312 | 0.625 | 0.314 | 0.373 | 0.226 | 1.532 |
| | $C_1$ (mg/g) | 0.141 | 0.162 | 0.114 | 0.289 | 0.115 | 1.126 |
| | $R^2$ | 0.821 | 0.932 | 0.873 | 0.797 | 0.897 | 0.814 |
| | SSE | 0.112 | 0.150 | 0.076 | 0.316 | 0.053 | 4.772 |
| | $k_{P2}$ (mg/(g min$^{1/2}$)) | 0.013 | 0.007 | 0.002 | -0.003 | 0.028 | 0.011 |
| | $C_2$ (mg/g) | 1.375 | 2.024 | 0.961 | 1.559 | 0.803 | 6.293 |
| | $R^2$ | 0.659 | 0.732 | 0.887 | 0.797 | 0.909 | 0.576 |
| | SSE | $4.831 \times 10^{-4}$ | $1.187 \times 10^{-4}$ | $3.174 \times 10^{-6}$ | $1.189 \times 10^{-5}$ | $4.565 \times 10^{-4}$ | $4.996 \times 10^{-4}$ |

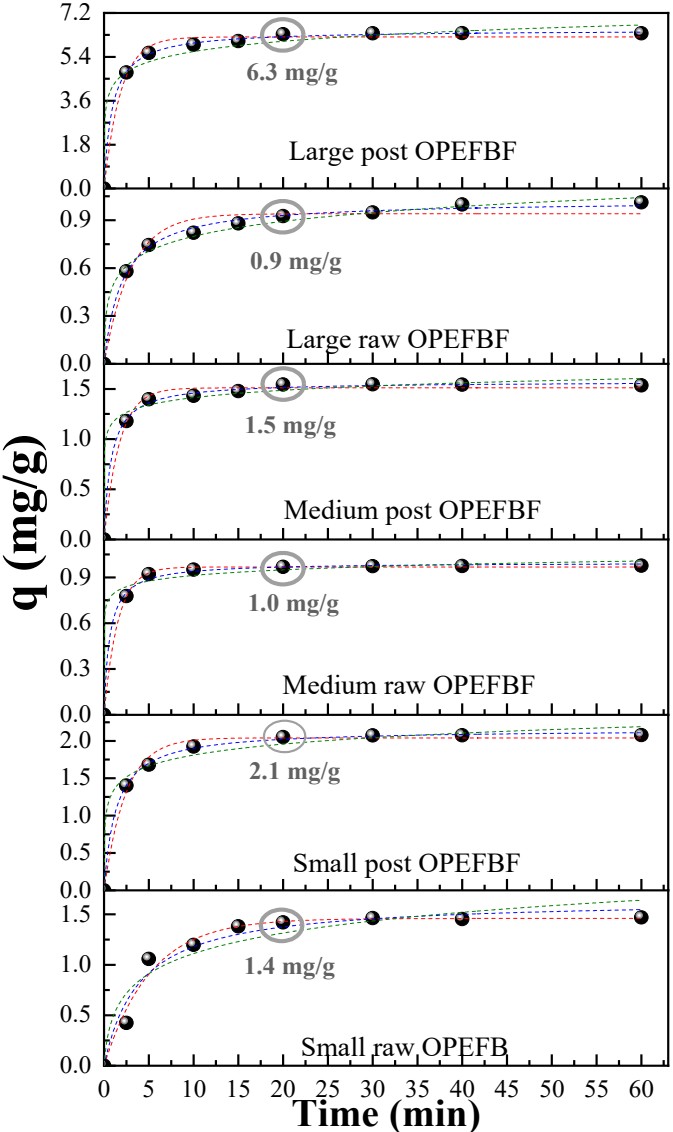

**Figure 6.** Kinetic models for MB (concentration = 5 mg/L) adsorption in raw/post OPEFBF. Black points = experimental data, red line = pseudo-first-order model (PFO), blue line = pseudo-second-order model (PSO), green line = Elovich model (E).

In Table 2, the fitting parameters for the intraparticle diffusion model are also provided. This model can be valuable for identifying reaction pathways and adsorption mechanisms, and it also enables the prediction of the rate-controlling step in adsorption [67]. In the context of adsorption of a contaminant present in an aqueous medium, the transfer of the contaminant commonly occurs through processes of film diffusion or external diffusion, surface diffusion and pore diffusion, or a combination of these mechanisms. The presence or absence of these processes can be determined through the analysis of the qt vs. $t^{1/2}$ curve. For instance, if the curve passes through the origin, it indicates that the adsorption process is solely limited by intraparticle diffusion. However, in Figure 7, two linear segments are observed, suggesting that the adsorption process is under the control of a two-step mechanism. In the first stage, film diffusion occurs, where MB is transported from the liquid phase to the external surface of raw/post OPEFBF through the hydrodynamic boundary layer. Meanwhile, in the second stage, intraparticle diffusion occurs, meaning there is a slow migration of MB molecules from the exterior of raw/post OPEFBF to their pores.

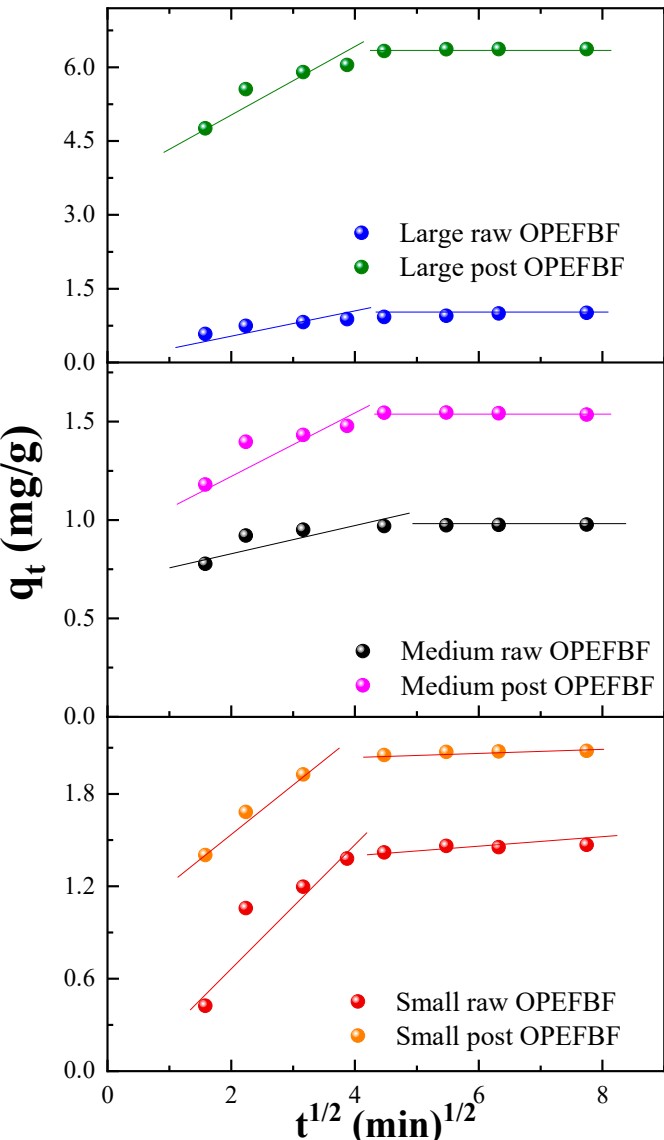

**Figure 7.** Intraparticle diffusion kinetics for adsorption of MB.

### 3.3.3. Adsorption Isotherms

Figure 8 shows that as the initial concentration of MB increases, the equilibrium adsorption capacity of raw/post OPEFBF also increases. To describe the relationship that occurs between the adsorbate and adsorbent when the concentration of the former is increased, the Langmuir and Freundlich models are widely used (parameters presented in Table 3). It is observed that the experimental data fit well with both models ($R^2$ between 0.933 and 0.996). However, when comparing the values of $R^2$, $\chi^2$ and SSE, it is noted that both raw/post OPEFBF materials exhibit a better fit to the Freundlich isotherm. The Langmuir model postulates a single-layer adsorption on uniform sites, implying a monolayer of adsorption with equal affinity for all adsorbed molecules. In contrast, the Freundlich model allows for multiple adsorptions on non-uniform sites without requiring surface uniformity [61–63,68].

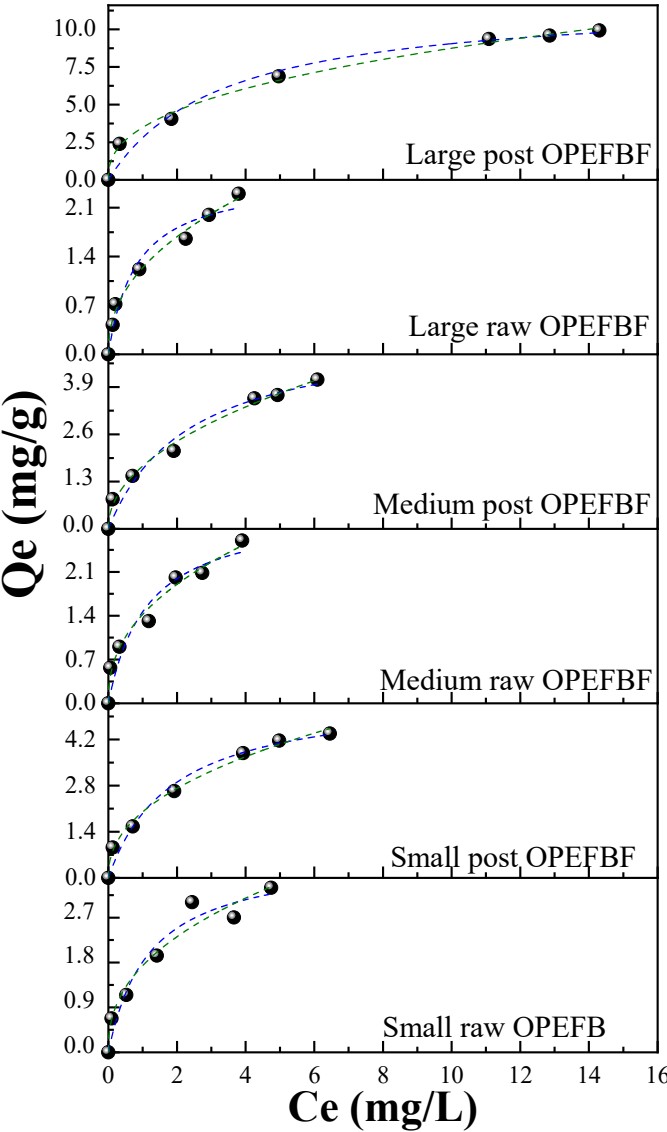

**Figure 8.** Isotherm models for MB (concentration = 5 mg/L) adsorption in raw/post OPEFBF. Black points = experimental data, blue line = Langmuir model and green line = Freundlich model.

**Table 3.** Parameters of isotherm models for MB adsorption.

| Model Type | Parameter (Units) | Small OPEFBF | | Medium OPEFBF | | Large OPEFBF | |
|---|---|---|---|---|---|---|---|
| | | Raw | Post | Raw | Post | Raw | Post |
| Langmuir | $q_{e\,exp.}$ (mg/g) | 3.008 | 3.787 | 2.938 | 3.590 | 1.655 | 9.590 |
| | $q_m$ (mg/g) | 3.969 | 5.611 | 2.721 | 5.535 | 2.746 | 12.022 |
| | $K_L$ [L/mg] | 0.841 | 0.537 | 0.881 | 0.421 | 1.227 | 0.305 |
| | $R^2$ | 0.956 | 0.979 | 0.933 | 0.970 | 0.965 | 0.980 |
| | $\chi^2$ | 0.084 | 0.074 | 0.068 | 0.089 | 0.958 | 0.375 |
| | SSE | 1.265 | 0.987 | 1.002 | 0.997 | 1.871 | 0.985 |
| Freundlich | $K_F$ [$(mg/g)^{1-1/n}$] | 1.737 | 2.003 | 1.4206 | 1.725 | 1.239 | 3.487 |
| | n | 2.404 | 2.283 | 2.375 | 2.087 | 2.263 | 2.503 |
| | $R^2$ | 0.962 | 0.994 | 0.982 | 0.994 | 0.992 | 0.996 |
| | $\chi^2$ | 0.071 | 0.019 | 0.018 | 0.019 | 0.007 | 0.068 |
| | SSE | 1.018 | 0.754 | 0.982 | 0.864 | 0.598 | 0.698 |

While the experimental data conform to the Freundlich model, the $q_e$ value in the Langmuir model provides insight into the adsorption capacity of OPEFBF. These values confirm what was previously mentioned: small raw OPEFBF exhibit the highest adsorption capacity (3.97 mg/g) compared to medium/large raw OPEFBF (2.72/2.75 mg/g), due to their higher specific surface [19]. Conversely, post OPEFBF materials display a higher adsorption capacity, with large post OPEFBF exhibiting the highest adsorption capacity (12.02 mg/g), which is attributed to the modification process experienced by the fibers during biofilter operation [60]. Additionally, the value of the Freundlich isotherm parameter (n), which is associated with adsorption intensity, falls within the range of 2.1 to 2.5, i.e., it lies in the Henry region: 1 < n < 10. This indicates favorable adsorption [61,68].

Some previous studies have also found a fit to the Freundlich isotherm, showing adsorption capacities that are comparable or even higher than those obtained in this study. For instance, activated carbon from *Millettia thonningii* seed pods ($q_e$ = 14.1 mg/g) [62] and natural Moroccan cactus (3.4 mg/g) [69] exhibited similar adsorption capacities. Nevertheless, even higher adsorption capacities have been observed in other cases, such as natural carbon ($q_e$ = 40.8 mg/g) [68] and the magnetic alginate/rice husk bio composite (344 mg/g) [70]. The greater adsorption capacity is often associated with thermal and chemical modifications applied to the adsorbent, which can impact the costs associated with the use of the adsorbent. For example, activated carbon is approximately 16 times more expensive than an adsorbent produced from $H_3PO_4$-modified berry leaves [71].

## 4. Conclusions

This study emphasizes the importance of effective wastewater treatment as a fundamental component of sustainable solutions, aligning with the sustainable development goals (SDGs). Furthermore, it underscores the relevance of addressing agro-industrial waste management on a global scale. By investigating the use of OPEFBF in different sizes as a filtering material in biofilters for the removal of organic matter and nutrients from wastewater and their subsequent application in MB removal through batch processes, we not only demonstrated that OPEFBF, whether in their raw or saturated state, are highly effective in contaminant removal (exceeding 80%), but also considered the revalorization of saturated materials. In contrast to other studies, we successfully modified these fibers through the action of microorganisms present in real wastewater, without the need for specific microorganisms like fungi and bacteria or their costly byproducts (e.g., enzymes). Based on the results obtained, our future prospects focus on conducting a controlled biological modification of OPEFBF using the same microorganisms present in wastewater, but under more controlled conditions. This will enable us to determine the optimal conditions for the fibers to acquire properties that make them an even more effective adsorbent. Ultimately, this study establishes the groundwork for future research that could have a significant impact on improving wastewater treatment processes and sustainable management of agro-industrial waste, thus contributing to the achievement of the SDGs and the pursuit of environmentally and economically sound solutions.

**Author Contributions:** Conceptualization, C.E.A.-N. and V.V.; methodology, E.D., C.E.A.-N., P.G., A.D.A. and A.D.; formal analysis, C.E.A.-N. and E.G.; investigation, C.E.A.-N. and V.V.; resources, V.V.; data curation, C.E.A.-N. and E.G.; writing—original draft preparation, C.E.A.-N. and E.G.; writing—review and editing, V.V. and C.V. visualization, C.E.A.-N. and V.V.; supervision, V.V.; project administration, V.V.; funding acquisition, C.E.A.-N. and V.V. All authors have read and agreed to the published version of the manuscript.

**Funding:** This research was funded by ESCUELA POLITÉCNICA NACIONAL, grant number PIS-22-23: "Composites bio-basados provenientes de residuos lignocelulósicos empleados en la remoción de metales pesados" and "The APC was funded by the Universidad de las Américas".

**Data Availability Statement:** Data are contained within the article.

**Acknowledgments:** The authors gratefully acknowledge the support provided by the Escuela Politécnica Nacional for the development of the project PIS-22-23: "Composites bio-basados provenientes de residuos lignocelulósicos empleados en la remoción de metales pesados".

**Conflicts of Interest:** The authors declare no conflict of interest.

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
