# Peer review of "From Renewable Biomass to Water Purification Systems: Oil Palm Empty Fruit Bunch as Bio-Adsorbent for Domestic Wastewater Remediation and Methylene Blue Removal"

_water, doi:10.3390/w15234116_

Round 1
Reviewer 1 Report
Comments and Suggestions for Authors
From renewable biomass to water purification systems: Oil palm empty fruit bunch as bio-adsorbent for domestic wastewater remediation and methylene blue removal.
A major revision is required.
1. This manuscript does not bring relevant contributions to the absorption field. First of all, methylene blue is a dye studied primarily in the literature because it is a model dye and not a dye that has a relevant contribution to the coloration of products. Besides that, other points, such as isotherms, follow the Freundlich model. What contribution does this bring to a review paper?
2. Lines 86-90- An activated carbon has no < 30 m²/g surface area. The carbon material described in these lines should be a biochar.
3. It is known that MB dye is not easily photodegraded. I did not see the need for the adsorption experiments to be performed in the dark.
4. SEM is not an analytical technique to define if a solid has good properties as an adsorbent. The data shown in Fig 3 does not prove the material is good. It is necessary to provide surface area (BET), pore size distribution curves, and total pore volume. Lines 325-326 are entirely wrong.
5. TGA analysis was poorly discussed in the text.
6. Please use "k" (lowercase) for constant kinetic rates. For equilibrium constants, please use "K" (CAPITAL LETTERS). See Table 2.
Comments on the Quality of English Language
The English language is ok.
Author Response
Dear Reviewer,
We would like to thank you for the comments and suggestions for the manuscript entitled: “From renewable biomass to water purification systems: Oil palm empty fruit bunch as bio-adsorbent for domestic wastewater remediation and methylene blue removal”. Following the suggestions received, we have modified the manuscript, and carefully revised the complete document. As a result, we believe that our manuscript has improved substantially, and conveys in a better manner some key points of our research.
To be more specific, we addressed the comments and suggestions received as follows:
- This manuscript does not bring relevant contributions to the absorption field. First of all, methylene blue is a dye studied primarily in the literature because it is a model dye and not a dye that has a relevant contribution to the coloration of products. Besides that, other points, such as isotherms, follow the Freundlich model. What contribution does this bring to a review paper?
We appreciate the reviewer's feedback; however, it's important to mention that we selected this dye because it's commonly used in studies of dye adsorption as a model dye, making it easier to compare different adsorbents. Its widespread availability, versatile chemical properties, several industrial applications, ease of detection and toxicity (Modi et al., 2022) contribute to its common use in research. Additionally, it's worth noting that in our region (Pelileo, Tungurahua province), methylene blue is extensively used in dyeing jeans, with an approximate monthly production of 1 million units for both local and international markets (Stadel, 2019). The textile industry in the area faces challenges related to the removal of this pollutant, and we aspire to contribute to solving this issue soon.
References:
- Modi, S., Yadav, V. K., Gacem, A., Ali, I. H., Dave, D., Khan, S. H., ... & Jeon, B. H. (2022). Recent and emerging trends in remediation of methylene blue dye from wastewater by using zinc oxide nanoparticles. Water, 14(11), 1749.
- Stadel, C. (2019). Horizontal and Vertical Archipelagoes of Agriculture and Rural Development in the Andean Realm. In Sustainability Assessment at the 21st century. IntechOpen.
Regarding the isotherms we have carefully reviewed our information and acknowledge that, by oversight, we initially omitted to include the data for fitting to the isotherm models in the text. However, we have rectified this omission and now provided detailed tables and graphs showing all the fitting parameters for Langmuir and Freundlich isotherms. We believe that this addition significantly strengthens the completeness of our work by offering a more comprehensive view of the results that can be useful in a review paper.
- Lines 86-90- An activated carbon has no < 30 m²/g surface area. The carbon material described in these lines should be a biochar.
It was specified that the mentioned specific surface area correspond to the raw OPEFB. Furthermore, this information was moved to lines 80-82 to clarify that it is the surface of the raw OPEFB.
- It is known that MB dye is not easily photodegraded. I did not see the need for the adsorption experiments to be performed in the dark.
While it is true that methylene blue (MB) dye is known for its relative photostability, conducting adsorption experiments in the dark was a deliberate choice to ensure the isolation of adsorption effects without potential interference from photodegradation. This is particularly important because the fiber had been previously used in a process where some byproduct could have been generated, and in the presence of light, it might lead to the photodegradation of MB.
- SEM is not an analytical technique to define if a solid has good properties as an adsorbent. The data shown in Fig 3 does not prove the material is good. It is necessary to provide surface area (BET), pore size distribution curves, and total pore volume. Lines 325-326 are entirely wrong.
We fully agree with the reviewer when mentioning that scanning electron microscopy is not an analytical technique to determine if a solid has good properties as an adsorbent. However, it's important to clarify that we don't assert this in our writing. Instead, we highlight the presence of functional groups identified through FTIR, and we have adjusted the wording to suggest, rather than affirm, that the irregularity of the surface may indicate OPEFB's potential as good adsorbents. Additionally, we want to inform the reviewer that we are currently working on another paper, which incorporates this information and hope to submit it soon. Unfortunately, progress has been hindered due to electricity rationing in our country, making continuous degassing challenging for OPEFB with its low surface area under current conditions.
- TGA analysis was poorly discussed in the text.
The discussion of the TGA analysis has been further elaborated, providing more in-depth insights into the presented results in Figure 2d. An initial mass loss below 150°C was observed in both raw and treated OPEFBF, attributed to water evaporation (volatile material). Between 150 and 410°C (moderately volatile material), a degradation process with a maximum decomposition temperature at approximately 310°C was identified. This phenomenon is linked to the degradation of hemicellulose, cellulose, and lignin, involving the removal of polyhydroxyl groups, accompanied by depolymerization and decomposition of macromolecules. This results in the formation of b-(1,6) anhydrous D-glucopyranose (levoglucosan), 2-furaldehyde (furfural), and volatile products of lower molecular weight. Additionally, at temperatures above 410°C, a residue rich in carbon and minerals persists [46,47]. Moreover, according to the DTG analysis, during microbial degradation, primarily of cellulose, hemicellulose, and lignin, various compounds were generated, including organic acids, aldehydes, ketones, partially degraded polysaccharides, phenolic compounds, and nitrogen-containing compounds [43,44]. The structure of these compounds can act as inhibitors for thermal reactions by altering the chemical balance or the availability of reaction sites. This interference could lead to a decrease in kinetics, as reflected in a second decomposition temperature at around 360°C [48].
- Please use "k" (lowercase) for constant kinetic rates. For equilibrium constants, please use "K" (CAPITAL LETTERS). See Table 2.
Letters was changed according to the reviewer suggestion.
Best regards,
Cristina Almeida
Reviewer 2 Report
Comments and Suggestions for Authors
Almeida-Naranjo and Valle reported the use of oil palm empty fruit bunch with different sizes as a filter bed material in biofilters designed for the removal of organic matter and nutrients from wastewater. Moreover, the saturated fibers were used as the adsorbents for the adsorption of methylene blue in water. The raw and post fibers were characterized by various means and the analysis results were discussed very well. The removal efficiencies of COD, total nitrogen and phosphorus were studied in the biofilters. And, the adsorption process of saturated fibers was investigated via the study of kinetics and isotherm models, and the adsorption conditions were finally optimized. Moreover, structure-reactivity relationship was studied to some extent. The work in the submission is completed and comprehensive, the subject of the upgrading of biomass is interesting and meaningful towards wastewater treatment. So, a minor should be performed before the consideration of the acceptance.
Some questions and issues:
1. The abbreviation should be checked carefully throughout the submission.
2. Line 65-67, the sentence has a grammar error.
3. The functional groups should be labeled in the FTIR spectra.
4. Line 386-387, the sentence has a grammar error.
5. Line 391, the description is confused. What does the removal stand for?
6. Line 443, what does “the raw large” mean?
7. Line 486, where is the Figure 8?
Comments on the Quality of English Language
Please see the comments to authors.
Author Response
Dear Reviewer,
We would like to thank you for the comments and suggestions for the manuscript entitled: “From renewable biomass to water purification systems: Oil palm empty fruit bunch as bio-adsorbent for domestic wastewater remediation and methylene blue removal”. Following the suggestions received, we have modified the manuscript, and carefully revised the complete document. As a result, we believe that our manuscript has improved substantially, and conveys in a better manner some key points of our research.
To be more specific, we addressed the comments and suggestions received as follows:
- The abbreviation should be checked carefully throughout the submission.
The abbreviation was checked carefully throughout the document.
- Line 65-67, the sentence has a grammar error.
It was corrected.
- The functional groups should be labeled in the FTIR spectra.
The main functional groups were added.
- Line 386-387, the sentence has a grammar error.
It was corrected as following: Previous research [57] highlighted that a smaller fiber size generally provided a larg-er surface area for adsorption (more activated sites).
- Line 391, the description is confused. What does the removal stand for?
The information was corrected to enhance text comprehension. It was mentioned that, in contrast, bed height showed significant results. Tejada-Tovar et al. (2018) [56] indicate that as the bed height increases, the number of active sites also increases, resulting in improved organic matter and nutrient removal.
- Line 443, what does “the raw large” mean?
The information specified that it refers to large OPEFBFs.
- Line 486, where is the Figure 8?
Figure 8 and Table 3 were included.
Best regards,
Cristina Almeida
Reviewer 3 Report
Comments and Suggestions for Authors
This is a very interesting paper on the use of oil palm empty fruit bunch as bio-adsorbent for domestic-scale wastewater treatment. The manuscript is interesting and well-written. I’d like to recommend its publication after minor revisions. Hopefully, my comments below will become useful in improving this manuscript.
Line 35: Please define COD prior to first use.
Line 57: Instead of “energy dependence”, I’d call it energy consumption.
Line 57: Isn’t the unit supposed to be kWh/m3?
Lines 57 – 58: Is the cost of 0.11 USD/m3 a maintenance cost or investment cost, or the cost that is derived from both investment and maintenance? Also, such a cost depends heavily on the location of such plant (region, country). Please provide more information as well as citations for this specific information.
Line 102 – 103: Authors define MB abbreviation as “methyl blue”, but then in Figure 1, it’s “methylene blue”. All over the materials and methods section MB is used. So, what was the adsorbate? Methyl blue (C37H27N3Na2O9S3) or methylene blue (C16H18ClN3S)? These are two different compounds (see https://pubchem.ncbi.nlm.nih.gov/compound/Methylene-Blue and https://pubchem.ncbi.nlm.nih.gov/compound/14513727 )
Table 1: Some typos need to be corrected, i.e., “standard deviation” and “dissolved oxygen”.
Figure 2: Something is wrong with DTG. Within the range, where there is the highest mass loss (as seen on TG curve) DTG has the lowest values of %/min on the axis. In the discussion, the authors have identified the ranges of highest mass loss well, so I’m certain that the only problem is the DTG axis in Figure 2 (d). Authors can have a look at DTG curves in the following papers, as examples and for comparison purposes: https://doi.org/10.3390/en13082058 ; https://doi.org/10.1115/1.4039906 One of the materials contains sugars (BSG), whereas the other sewage sludge is a waste from wastewater treatment, and also contains dead microorganisms. Since in Your discussion of TGA, You refer to the accumulation of sugars and the influence of microorganisms on Your material – it could be helpful.
Author Response
Dear Reviewer,
We would like to thank you for the comments and suggestions for the manuscript entitled: “From renewable biomass to water purification systems: Oil palm empty fruit bunch as bio-adsorbent for domestic wastewater remediation and methylene blue removal”. Following the suggestions received, we have modified the manuscript, and carefully revised the complete document. As a result, we believe that our manuscript has improved substantially, and conveys in a better manner some key points of our research.
To be more specific, we addressed the comments and suggestions received as follows:
Line 35: Please define COD prior to first use.
The mean COD was included the first time it appeared.
Line 57: Instead of “energy dependence”, I’d call it energy consumption.
It was corrected.
Line 57: Isn’t the unit supposed to be kWh/m3?
It was corrected.
Lines 57 – 58: Is the cost of 0.11 USD/m3 a maintenance cost or investment cost, or the cost that is derived from both investment and maintenance? Also, such a cost depends heavily on the location of such plant (region, country). Please provide more information as well as citations for this specific information.
We add in the manuscript the operating cost of a conventional activated sludge water treatment. This is derived from an economic and technical analysis, where the capital cost and the operating cost of two types of wastewater treatment, including activated sludge treatment, were calculated. The study was carried out in Canada, and it is presented in (Gao., et al 2021).
Gao, T., Xiao, K., Zhang, J., Xue, W., Wei, C., Zhang, X., Liang, S., Wang, X., & Huang, X. (2021). Techno-economic characteristics of wastewater treatment plants retrofitted from the conventional activated sludge process to the membrane bioreactor process. Frontiers of Environmental Science & Engineering, 16(4), 49. https://doi.org/10.1007/s11783-021-1483-6
Line 102 – 103: Authors define MB abbreviation as “methyl blue”, but then in Figure 1, it’s “methylene blue”. All over the materials and methods section MB is used. So, what was the adsorbate? Methyl blue (C37H27N3Na2O9S3) or methylene blue (C16H18ClN3S)? These are two different compounds (see https://pubchem.ncbi.nlm.nih.gov/compound/Methylene-Blue and https://pubchem.ncbi.nlm.nih.gov/compound/14513727 )
We worked with methylene blue, and the abbreviation for it was corrected throughout the document. Furthermore, a distinction was made between methylene blue (MB) and methyl blue.
Table 1: Some typos need to be corrected, i.e., “standard deviation” and “dissolved oxygen”.
The information of Table 1 was corrected.
Figure 2: Something is wrong with DTG. Within the range, where there is the highest mass loss (as seen on TG curve) DTG has the lowest values of %/min on the axis. In the discussion, the authors have identified the ranges of highest mass loss well, so I’m certain that the only problem is the DTG axis in Figure 2 (d). Authors can have a look at DTG curves in the following papers, as examples and for comparison purposes: https://doi.org/10.3390/en13082058 ; https://doi.org/10.1115/1.4039906 One of the materials contains sugars (BSG), whereas the other sewage sludge is a waste from wastewater treatment, and also contains dead microorganisms. Since in Your discussion of TGA, You refer to the accumulation of sugars and the influence of microorganisms on Your material – it could be helpful.
The thermogram was corrected, and the discussion was expanded, taking into account the information found in the second paper suggested by the reviewer, as following: An initial mass loss below 150°C was observed in both raw and treated OPEFBF, attributed to water evaporation (volatile material). Between 150 and 410°C (moderately volatile material), a degradation process with a maximum decomposition temperature at approximately 310°C was identified. This phenomenon is linked to the degradation of hemicellulose, cellulose, and lignin, involving the removal of polyhydroxyl groups, accompanied by depolymerization and decomposition of macromolecules. This results in the formation of b-(1,6) anhydrous D-glucopyranose (levoglucosan), 2-furaldehyde (furfural), and volatile products of lower molecular weight. Additionally, at temperatures above 410°C, a residue rich in carbon and minerals persists [46,47]. Moreover, according to the DTG analysis, during microbial degradation, primarily of cellulose, hemicellulose, and lignin, various compounds were generated, including organic acids, aldehydes, ketones, partially degraded polysaccharides, phenolic compounds, and nitrogen-containing compounds [43,44]. The structure of these compounds can act as inhibitors for thermal reactions by altering the chemical balance or the availability of reaction sites. This interference could lead to a decrease in kinetics, as reflected in a second decomposition temperature at around 360°C [48].
Best regards,
Cristina Almeida
Round 2
Reviewer 1 Report
Comments and Suggestions for Authors
This paper could be accepted.